# Performance Analysis of Two-Zone-Based Registration System with Timer in Wireless Communication Networks

Hee-Seon Jang [1] and Jang-Hyun Baek [2],*

1 Department of Convergence Software, Pyeongtaek University, 3825, Seodong-daero, Pyeongtaek-si 17869, Gyeonggi-do, Republic of Korea; hsjang@ptu.ac.kr
2 Department of Industrial & Information Systems Engineering and the RCIT, Jeonbuk National University, 567, Baekje-daero, Deokjin-gu, Jeonju-si 54896, Jeollabuk-do, Republic of Korea
* Correspondence: jbaek@jbnu.ac.kr; Tel.: +82-63-270-2330

**Abstract:** Numerous studies have been conducted on wireless communication networks to reduce the costs associated with location registration and paging traffic caused by the movement of user equipment (UE). Among them, a zone-based registration scheme is commonly used due to its convenience of implementation. In a zone-based scheme, a set of non-overlapping cells are managed as a single zone called a registration area (RA). The UE requests registration each time it enters a new RA. The most significant drawback of the 1Z system (an RA consisting of one zone) is the degradation in the service quality due to the registration traffic that frequently occurs at the RA boundaries. To overcome this drawback, a 2Z system that can manage two zones with one RA has been proposed. However, in the 2Z system, the paging costs for the UE that has returned to the previous zone increase, which can significantly degrade the performance compared to the 1Z system if the call-to-mobility ratio (CMR) is large or if the probability of returning to the previous zone is small. In this study, a new 2Z_*Timer* scheme is proposed to enhance the performance of the 2Z system. This method involves initiating a *Timer* for the previously visited zone when the UE enters a new zone, making it possible to retain the information of the previous zone for a specified threshold period. Simulations were conducted using flowchart-based RAPTOR software to compare its performance to those of the 1Z and 2Z systems. The results showed that the 2Z_Timer system effectively reduced the paging costs, even when the CMR was high or the probability of returning to the previous zone was low. Numerical results for various *Timer* thresholds showed that the 2Z_Timer system could lead to cost reductions of 10.6% and 28.6% compared to the 1Z and 2Z systems, respectively.

**Keywords:** mobility management; location registration; paging; zone-based registration; timer-based registration; two-zone registration; RAPTOR simulation

## 1. Introduction

In a wireless communication network, the continuous tracking and management of the user equipment (UE)'s current location information is crucial to successfully establish incoming calls for the UE. This function is referred to as mobility management. It was found that mobility management consumes a significant amount of wireless frequency resources [1–3]. In the case of 4G LTE, 34% of all traffic is related to mobility management [4]. An efficient data traffic service is possible only when it is effectively managed [4–6]. Mobility management includes two essential operations: location registration and paging [1,3,7,8]. Location registration is the procedure in which the UE reports its current location information to the network every time it enters a new registration area (RA). The network's database then updates this information. For the efficient processing of location registration, each RA is composed of one or more zones, and each zone is structured as a set of cells that do not overlap with each other. Here, a 'cell' represents the area served by a single wireless base station. Paging is the process of sending paging messages to all cells included in the current RA of the UE stored in the network database when an incoming call is received. Therefore, the number of cells (or zones) included

in the RA can affect the costs of the UE registration and paging-signal traffic (i.e., wireless signaling costs). In other words, as the RA's size increases, the registration cost decreases and the paging cost increases. Ultimately, the RA's size and the associated costs of registration and paging have a trade-off relationship [9–11].

To date, many studies have been carried out to enhance the utilization of wireless frequency resources for mobility management, as shown in Table 1. These studies can be categorized into zone-based [10–13], distance-based [14,15], movement-based [16,17], timer-based [18,19], and mobility prediction-based [20]. In particular, new mobility management schemes for various moving devices beyond 5G and 6G have been recently discussed [21–23]. In Table 1, assuming a square-shaped RA, the registration occurrence points in each scheme are compared when the UE moves from *s* to *t*. The major advantages and limitations of the schemes are also examined. The performance of each scheme depends on the UE's movement characteristics, such as the call-to-mobility ratio (CMR) and the probability (*q*) of returning to the previously visited RA. Here, the CMR indicates the ratio of the number of calls to the number of times the UE passes through the RA boundary during the unit time. That is, the larger the CMR, the more frequently the UE has the characteristic of staying in the RA for a long time and trying to initiate calls. Typically, if a 1Z scheme with one RA and a 2Z scheme with two RAs are compared to each other, the larger the CMR and the smaller the *q*, the worse the performance of the 2Z scheme compared to the 1Z scheme. Recently, a new algorithm has been proposed to efficiently manage registration by predicting the UE's moving direction in advance. In addition, new schemes to utilize artificial intelligence machine learning algorithms using training data on the UE mobility characteristics have been proposed [22,23].

For the convenience of implementation, most standards [1,2,19] recommend a 1Z scheme with one zone in an RA. Lin [10] first proposed a 2Z scheme with two zones in one RA and showed that the greater the probability of returning to the previous RA, the better the performance of the 2Z scheme. In addition, a 3Z scheme with three zones within one RA via expanding the 2Z scheme has been proposed [11,12]. The performances of the three schemes for various parameters were compared [13]. As mentioned, despite many research efforts, in practice, most wireless communication networks adopt a zone-based registration scheme due to its implementation convenience. That is, the UE requests registration each time it crosses the RA's boundary. When an incoming call arrives, paging is processed for all cells within that RA. The most significant drawback of the 1Z system, which operates one zone within an RA, is the handling of the frequent registration traffic (known as the ping-pong effect) at the RA boundaries, resulting in service quality degradation for other subscribers [10,24–28]. To address this issue, a scheme involving the operation of two zones within an RA (the 2Z system) has been proposed [10,13]. In the 2Z system, there is no registration cost for the UE's return to the previous zone. However, during the call setup, paging needs to be executed in both the previous and new zones, leading to twice the paging cost compared to the 1Z system.

This study was based on the 2Z system. Additionally, this study suggests initiating a *Timer* when the UE enters a new zone, allowing the UE to store information from the previous zone for a specific time (the *Timer* threshold ($T_h$)). This approach can reduce the paging costs for UE that has completed additional registration due to *Timer* expiration, including that which has entered a new zone. To evaluate the performance of the proposed 2Z_Timer system and compare it with the existing 1Z and 2Z systems, a simulation model was developed using visual-flowchart-based RAPTOR software of Avalonia Edition [29].

The remaining sections of this paper are structured as follows. Section 2 explains the operational methods of the 1Z, 2Z, and 2Z_Timer systems. It also describes the assumptions necessary for the convenience of simulation, where a large wireless communication service network is downscaled. Section 3 outlines the simulation model and explains additional assumptions for the call arrival time and dwell time within a zone. Section 4 presents simulation results using real-operational-environment parameters and compares the performances of the three systems. Finally, conclusions are drawn and additional research directions are presented in Section 5.

**Table 1.** Summary of related works (●: registration; ○: no registration).

| Schemes | Methodologies | Advantages | Limitations | References |
|---|---|---|---|---|
| Zone-based | One zone-based (1Z)<br><br>Registration whenever RA boundary is passed | Easy to implement | High registration traffic at RA boundaries | [11,12] |
| | Two zone-based (2Z)<br><br>No registration when passing two RA boundaries | Reduced registrations at RA boundaries | Paging traffic resulting from paging two RAs | [10,13] |
| Distance-based | Registration if UE moves above threshold (=2) of distance | Reduced registration traffic moving only in a specific area | High paging traffic as threshold increases | [14,15] |
| Movement-based | Registration every time UE passes the RA boundary moving threshold (=2) | Reduced registration traffic for UEs that do not move much | Difficult to implement, including moving-threshold value setting | [16,17] |

**Table 1.** *Cont.*

| Schemes | Methodologies | Advantages | Limitations | References |
|---|---|---|---|---|
| Timer-based |  Registration after the *Timer* threshold exceeds *t* | Accurately locates UEs not moving for a long time | No significant difference in performance from 1Z scheme | [18,19] |
| Mobility prediction-based |  Dynamic RA's configuration using the probability of returning to the previous RA | Efficient registration management by predicting UE's moving direction | Difficult to predict the *q* | [20] |

## 2. Zone-Based Registration Scheme with Timer

Most wireless communication networks use a zone-based registration method due to its convenience of implementation. When there is only one zone within an RA, it is referred to as a 1Z system. When managing two zones (i.e., the previously visited zone and the newly entered zone), it is called a 2Z system. The approach proposed in this study, which involves using a *Timer* for the previous zone, is named the 2Z_Timer system. For the mobility modeling of the UE, the following assumptions are adopted:

(i)  An area served by each wireless base station is called a cell, and the cell is formed in a square shape;

(ii)  Each zone consists of $n$ cells. In a 1Z system, an RA is composed of one zone, whereas in a 2Z system, an RA is composed of two zones. Therefore, in a 1Z system, a new registration must be processed every time the zone's boundary is passed. However, in a 2Z system, registration is not required when the UE returns to the previously visited zone;

(iii)  The probability of the UE returning to the previous zone is $q$, and the probability of the UE moving in any one of the remaining three directions is equally $(1 - q)/3$;

(iv)  The paging delay is 1. That is, when setting the incoming call, $n$ cells are simultaneously paged in the 1Z system, or $2 \times n$ cells in the 2Z system, to find the corresponding UE.

### 2.1. One-Zone System (1Z System)

Figure 1 shows the operating characteristics according to the UE's movement in the 1Z system when $n = 4$. The UE moves to the previous zone with the $q$. The 1Z system

processes registration every time the UE moves across the RA's (or zone's) boundary. That is, new registration traffic is generated whenever the UE moves to cells that belong to a different RA. As such, the phenomenon of the deteriorating service quality of other UE due to the registration traffic caused by the UE's movement that frequently crosses the RA boundary is called the ping-pong effect [10,24,27].

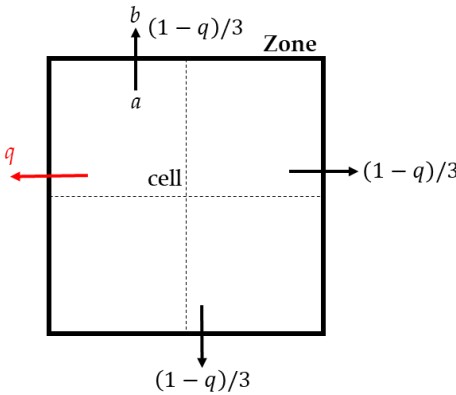

$q$: probability of returning to previously visited zone

**Figure 1.** A 1Z system (one zone in a registration area).

It is necessary to develop a simulation model by downsizing the overall coverage area served by the wireless communication network. As shown in Figure 1, when the UE moves from $a$ to $b$, a registration procedure is required. After executing the registration, the UE stays in the zone again. In other words, it is assumed that the simulation is carried out by configuring only one zone. When the UE selects one of the four directions and moves, it stays in the same zone again. In the 1Z system, registration is carried out every time the UE enters a new RA. Thus, the network can accurately know the UE's RA information. Thus, the UE can be found by paging $n$ cells included in the RA during a call connection.

*2.2. Two-Zone System (2Z System)*

There is no registration cost when the UE moves between an old zone and a newly registered zone (new zone) in the 2Z system, unlike in the 1Z system. As shown in Figure 2, after the registration is terminated, if the UE moves from $c$ to $d$, then the network database stores two zones (new and old zones). According to the UE's movement, the solid line indicates a case in which registration is executed and the dotted line describes no registration. If the UE proceeds from $e$ to $f$ and returns to the old zone, registration is not required. Additionally, when the UE moves back to the new zone with a probability ($q$), there is no registration cost. This can greatly reduce the registration costs for UE that frequently crosses the RA boundary between old and new zones.

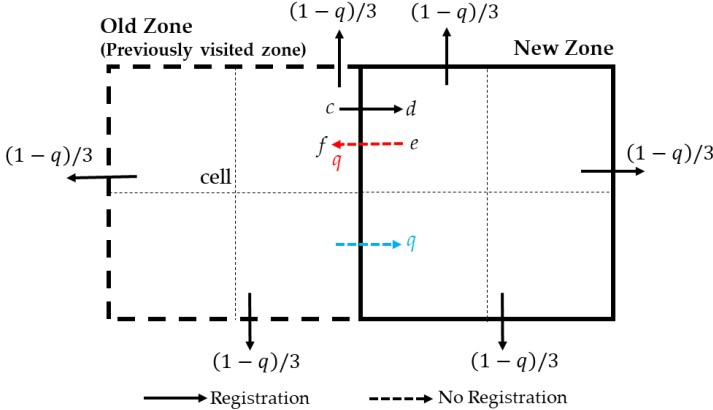

**Figure 2.** A 2Z system (two zones in a registration area).

In a 2Z system, simulations are executed considering only two zones: new and old zones. Here, if the UE moves to another zone and requests registration, it stays in the new zone. If it returns to the old zone, it is considered to be in the old zone. In Figure 2, assuming a paging delay of 1, which means that paging messages must be sent in a single attempt when the incoming call arrives, eight cells ($2n = 8$) need to be paged to successfully establish the call in a 2Z system. In other words, because the UE returns to the old zone without requesting registration, the network does not know whether the UE is in the new zone or back in the old zone. Thus, in a 2Z system, $2n$ cells included in new and old zones must be paged at all times when setting the incoming call.

### 2.3. Two-Zone System with Timer (2Z_Timer System)

The 2Z_Timer system is based on the 2Z system. Additionally, when the UE proceeds to the new zone, it drives the *Timer* for the previous zone. The network checks whether the *Timer* for the UE returning to the old zone has expired. If the *Timer* has expired (that is, *Timer* $\geq$ *Threshold* = $T_h$), a new registration is performed. Therefore, additional registration costs are required when the *Timer* expires after returning to the old zone. However, compared to those of the 2Z system, the paging costs of the 2Z_Timer system can be reduced.

Figure 3 shows the operation of a 2Z_Timer system. Similar to a 2Z system, whenever the UE crosses between old and new zones, no registration costs are incurred. The moving probability between zones is the same as shown in Figure 2. In Figure 3, the (★) indicates the case of an incoming-call occurrence in each zone. The 2Z_Timer is operated as follows to connect an incoming call:

(i) To initiate the incoming call generated, $n$ cells are paged while the UE stays in the new zone. As in (★1), if the UE moves from $c$ to $d$ and an incoming call occurs, the call can be successfully established by paging $n = 4$ cells;

(ii) If the *Timer* expires after re-entering the old zone, the UE's RA information is updated at an additional registration cost, and $n$ cells are paged for an incoming-call setup staying in that zone. In (★2), after the *Timer* expires and the registration is processed, four cells are paged while residing in the old zone;

(iii) In contrast, if the *Timer* does not expire after relocating in the old zone, $2n$ cells need to be paged, in the same way as in the 2Z system. As shown in (★3), if the *Timer* has not expired, $2n = 8$ cells are paged.

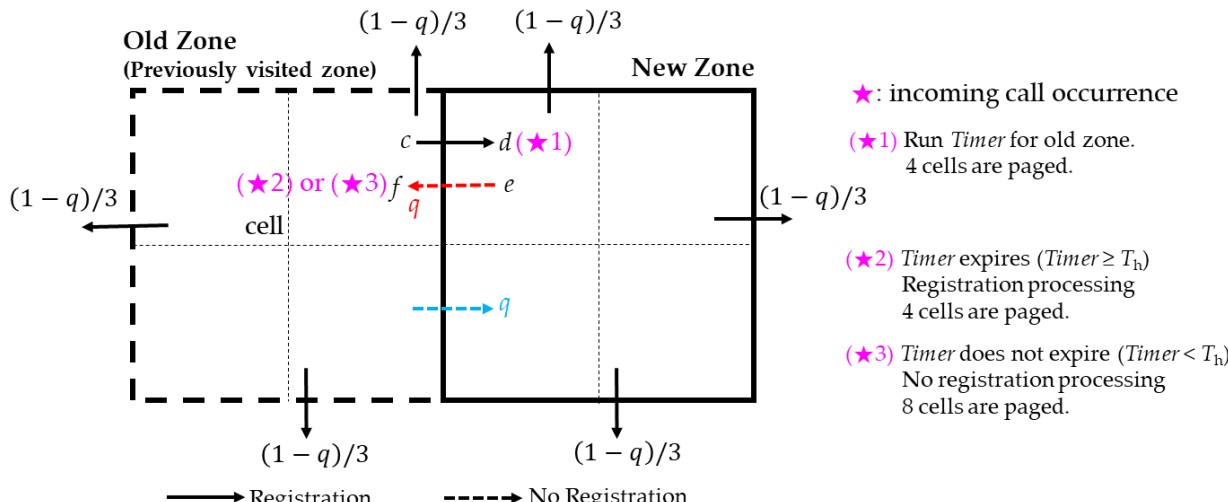

**Figure 3.** A 2Z system with *Timer* (two zones in a registration area and the use of *Timer*).

Table 2 shows comparisons of the registration and paging trials for the three systems while the UE travels from $s$ to $t$. This example shows a situation in which the UE connects

two incoming calls (★) as it moves and returns to the previous RA. In the 1Z system, three registrations are treated every time the UE crosses the RA's boundary. A total of eight cells are paged for two incoming calls while returning to the *t* point. In the 2Z system, when the UE returns to the previous RA, registration is not executed (the number of registrations is two) and a total of 16 cells are paged while setting up calls. In contrast, in the 2Z_Timer system, the *Timer* is first checked for the UE that has returned to the previous RA. If the *Timer* is expired (Case 1), an additional registration is needed (a total of three registrations) and eight cells are paged for two calls. When the *Timer* does not expire (Case 2), registration is not required (the number of registrations is two) and a total of 12 cells are paged by combining 4 cells for the first call and 8 cells for the second call.

**Table 2.** Operations of three systems (●: registration; ○: no registration, ★: incoming call, *n* = 4).

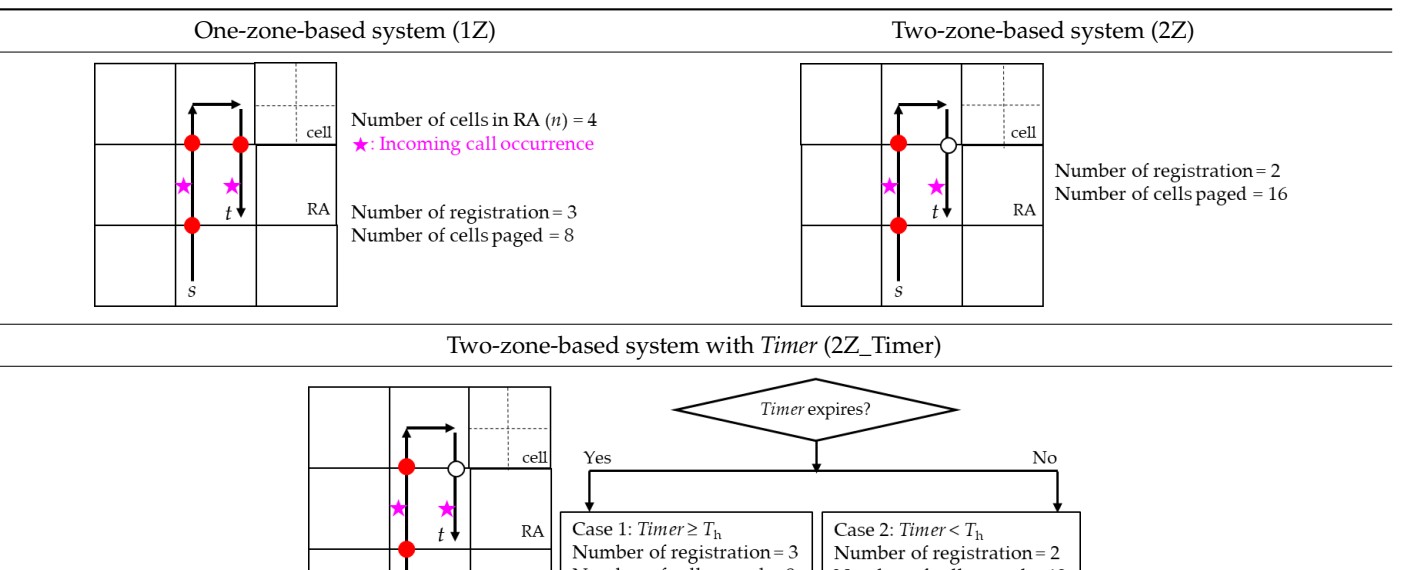

Therefore, the 2Z_Timer system takes advantage of the 2Z system without performing registration when the UE returns to the previous RA. In addition, the paging costs can be reduced compared to the 2Z system by additional registration due to *Timer* expiration. In the following section, a simulation model is described and the performances of the three systems are compared. The system performance can be greatly influenced by various parameters, such as the *Timer* threshold ($T_{\mathrm{h}}$), CMR, *q*, and number of cells (*n*) in the RA.

### 3. Simulation Model

Figure 4 shows a flowchart of the simulation. For the simulation, the following was additionally assumed:

(i) The number of calls generated in a unit time (h) follows a Poisson distribution with a mean of $\lambda$. The rates of the originating ($\lambda_{out}$)- and incoming ($\lambda_{in}$)-call occurrences are equal. That is, $\lambda = \lambda_{out} + \lambda_{in}$. Thus, the intervals between call arrivals are distributed exponentially with an average of $1/\lambda$ [11–14,30];

(ii) According to the implicit registration function (IRF) provided by a wireless communication network [1,12,19], the UE's RA information can be updated without an additional registration cost after the originating- or incoming-call setup;

(iii) The UE's residence time in a zone follows a gamma distribution with the parameters shape = $\alpha$ and scale = $\beta$. The average dwell time is $\mathrm{E}[T_m] = \frac{1}{\mu} = \alpha\beta$ [30].

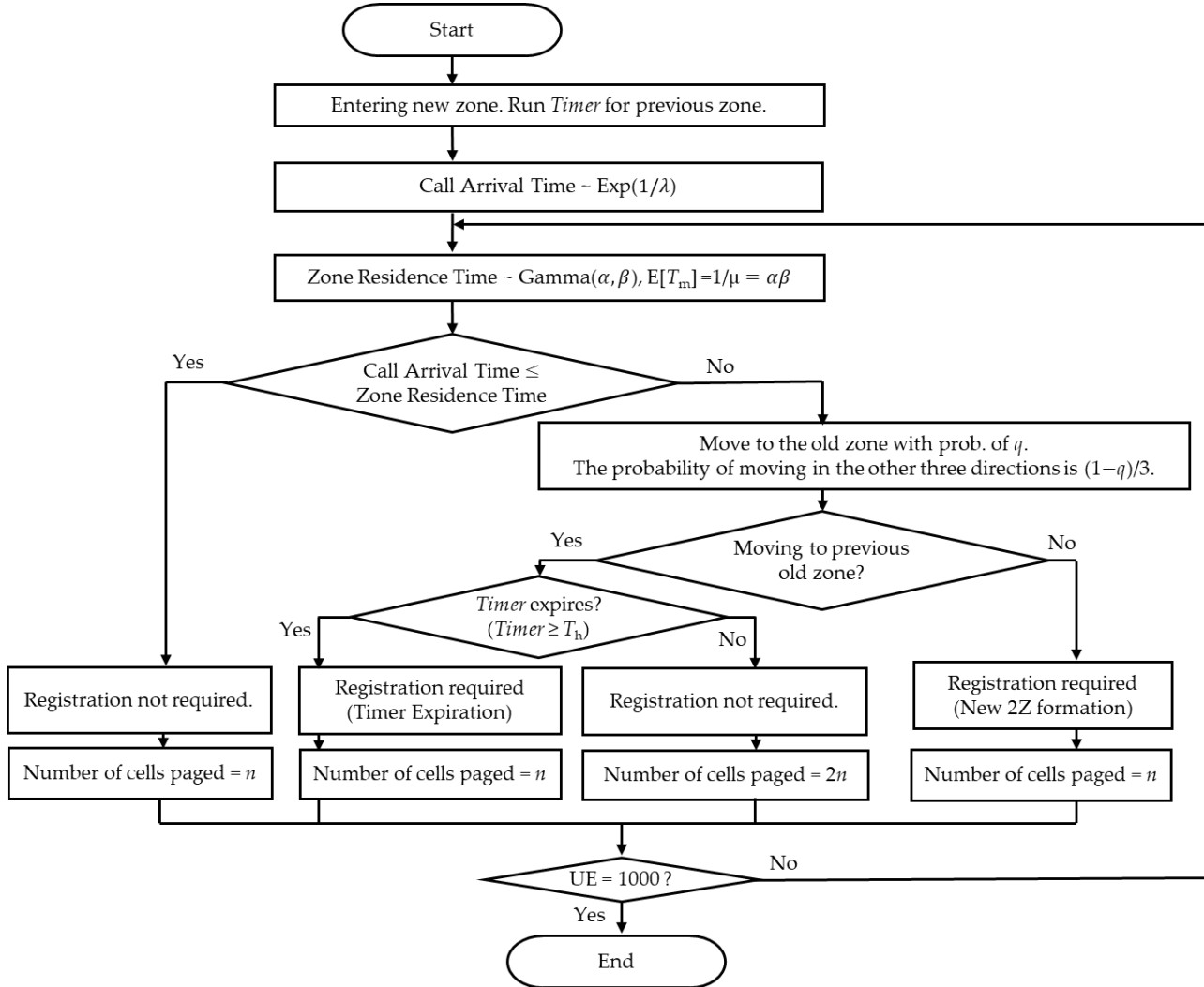

**Figure 4.** Simulation model.

The simulation was performed using RAPTOR [29], a free, open, and visual-flowchart-based software program. Using the RAPTOR menu, the flowchart-based design can be easily converted into the Java or C++ programming languages, enabling more detailed algorithm implementation and testing. Ten simulations were carried out, with 1000 units of UE using different seed numbers to calculate the registration and paging costs for a total of 10,000 units of UE, as follows.

### 3.1. Location Registration

In the initial state, it was assumed that the UE had moved from another zone to a new zone. In the case of the 2Z_Timer system, the Timer for the previous zone was started. If the call arrival time to the UE was less than or equal to the staying time in the zone, registration was not performed. For the incoming call, $n$ cells were paged. When the UE moved to another zone before the call arrived, it proceeded to the previous zone with a probability of $q$ and to one of the remaining three zones with a probability of $(1 - q)/3$. The probability of movement between zones is presented in Figures 1 and 2.

In the 1Z system, registration must be executed whenever the UE enters a new RA. In contrast, if the UE re-enters a zone within the previous zone, registration is not required in the 2Z or 2Z_Timer systems. In the 2Z_Timer system, when the UE moves to the previous zone, the *Timer* value operated earlier is checked. If the *Timer* has expired (*Timer* $\geq T_h$), a new registration procedure is completed, and $n$ cells are paged in the case of an incoming

call generated while staying in that zone. If the *Timer* has not expired, $2n$ cells need to be paged for an incoming call. The paging scheme for each system is explained below in detail.

*3.2. Paging*

The paging cost is calculated for each system as follows:

(i) In a 1Z system, when an incoming call arrives, the corresponding UE for a call connection can be accurately found by paging $n$ cells at the same time;

(ii) In a 2Z system, $2n$ cells must be paged simultaneously to make the incoming call successful under a paging delay of 1;

(iii) Unlike the 1Z and 2Z systems, the 2Z_Timer system pages $n$ or $2n$ cells depending on whether the *Timer* has expired:

    (a) To ensure an incoming-call initiation for a UE that has entered a new zone before returning to the previous zone, $n$ cells are paged;

    (b) When the *Timer* of the UE returning to the previous zone has not expired, $2n$ cells are paged. If the *Timer* has expired, a new registration procedure (registration via *Timer* expiration) is processed, and $n$ cells are paged to find the UE before entering another zone.

The system parameters required to perform the simulation were referred to in the following studies:

(i) For the most basic parameter, assuming CMR = 1, the performance of the 2Z_Timer system was analyzed and compared with those of the other systems. A CMR = 1 represents a situation in which the UE attempts one call on average while moving to another RA, which was assumed in most studies [10–16] to compare the system performances. Note that when $\lambda = 1$ and $E[T_m] = 1$, CMR = 1 was obtained, on average;

(ii) In zone-based registration, the 3GPP [1,19] recommends that UE staying in one RA for a long time be periodically registered with a Timer threshold. Here, the system can select one of the zone timer lengths (1, 2, 5, 10, 20, 30, 45, or 60 min);

(iii) Assuming a square RA, the probability of moving in one of the four directions was 1/4 [10,12–14]. The simulation was executed on the assumptions that the UE did not return to the previous RA ($q = 0.1$) and that the UE frequently returned to the previous RA ($q = 0.5$);

(iv) The number of cells included in an RA was analyzed for environments of 4, 9, and 16, assuming square-shaped cells [12–15];

(v) The unit registration cost was assumed to be 10 times the paging cost per cell for an incoming call. This was assumed in several studies [1,2,10,19].

## 4. Numerical Results

The performances of the three systems varied depending on various factors. The simulation was conducted to obtain the total registration cost and paging cost of each UE during the unit time. The performances were compared for variable factors, such as the call-to-mobility ratio (CMR = $\frac{\lambda}{\mu}$), $T_h$, $q$, and $n$.

Considering real operational environments, the following fundamental parameters were used:

$n = 9$: the number of cells in one zone;

$\lambda = \lambda_{out} + \lambda_{in} = 0.5 + 0.5 = 1$: the average call occurrence rate over the unit time;

$V = 1$: the unit paging cost per one cell for an arriving incoming call [1,19];

$U = 10$: the unit processing cost for one registration [1,19].

The unit registration cost was incurred when the UE relocated to a new RA. The unit cost associated with the *Timer* expiration was assumed to be equal to $U = 10$ [1,2,10,19]. Note that the outgoing-call and incoming-call occurrence rates during the unit time were the same. After call processing, the network could update the UE's zone information without

additional registration costs by the IRF. As for the *Timer* threshold, the representative $T_h$ = 1, 2, 5, 20, and 60 min recommended by the 3GPP [19] were considered.

Figure 5 shows the total cost versus the CMR when $T_h$ = 20 min and *q* = 0.4. For example, CMR = 1 refers to a situation in which a UE connects one call while staying in a zone for an hour on average. The 2Z_Timer system checked the *Timer* for the UE that returned to the old zone and performed registration if the *Timer* was more than 20 min. It did not require registration if the *Timer* was less than 20 min. In the figure, the solid line shows the total (registration + paging) cost (left side), and the dotted line indicates the relative reduction (reduction ratio (%), right side) in the cost. Also, in the dotted line, the O-shape represents the relative reduction ratio of the 2Z_Timer system compared to the 2Z system, and the X-shape represents the reduction ratio of the 2Z_Timer system compared to the 1Z system. If the reduction ratio has a negative value, the cost of the 2Z_Timer scheme is relatively large. If it has a positive value, the cost of the 2Z_Timer scheme is smaller than those of the other schemes.

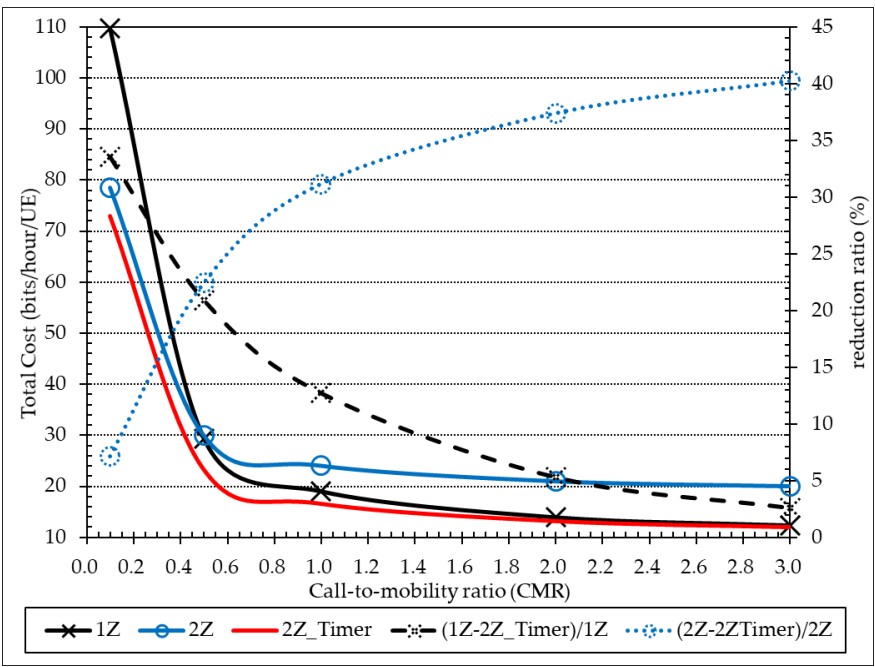

**Figure 5.** Total cost and reduction ratio versus CMR (call-to-mobility ratio).

When the CMR increased, the 1Z system outperformed the 2Z system. The reason was that the larger the CMR (that is, the longer the UE's residence time in the zone than the call arrival time interval), the less likely the UE is to return to the previous zone, and thus the greater total cost for the 2Z system than for the 1Z system. In the case of the 2Z_Timer system, it consistently showed smaller total costs than the other systems. The higher the CMR of the UE, the longer it remained in the zone, enabling the more accurate identification of its location, ultimately resulting in a lower paging cost for the 2Z_Timer system. The results showed that the total cost of the 2Z_Timer system was reduced by 15.0% and 27.7% compared to those of the 1Z and 2Z systems, respectively.

Figures 6 and 7 show the registration and paging costs according to the *Timer* thresholds ($T_h$) under CMR = 1 and *q* = 0.4. Note that for the 1Z and 2Z systems, the registration and paging costs were not influenced by the $T_h$. Thus, the cost remained constant regardless of the $T_h$. However, in the 2Z_Timer system, when the $T_h$ increased, the registration cost decreased while the paging cost increased slightly. Accordingly, it was observed that the registration cost of the 2Z_Timer system was greater than that of the 2Z system. However, the difference decreased as the $T_h$ increased. Note that if the $T_h$ was more than a certain amount of time ($T_h$ = 20 min when CMR = 1), the registration cost no longer decreased.

In addition, the paging cost did not change significantly. On average, the registration cost of the 2Z_Timer system was increased by 20.3% compared to that of the 2Z system, whereas it was decreased by 29.4% compared to that of the 1Z system. As shown in Figure 7, regardless of the $T_h$, when installing the incoming call, 9 cells of the 1Z system and 18 cells of the 2Z system were paged. In contrast, in the 2Z_Timer system, a substantial reduction in the paging cost was achieved by paging only nine cells for the UE entering a new zone and the UE that returned to the previous zone (and performed registration due to the *Timer* expiration). Based on these results, it was evident that the paging cost of the 2Z_Timer system was always less than that of the 2Z system. The paging cost of the 2Z_Timer system was slightly higher than that of the 1Z system. The paging cost of the 2Z_Timer system decreased by 44.7% on average compared to that of the 2Z system, and it increased by 10.6% compared to that of the 1Z system.

The total cost was calculated by adding the registration and paging costs. As shown in Figure 8, when the paging delay was 1 (that is, when an immediate connection was required during an incoming-call setup), the total cost of the 2Z system was always greater than that of the 1Z system. The 2Z_Timer system always performed better than the other systems. This was because, in the 2Z_Timer system, the paging costs could be effectively reduced by conducting registration only when the *Timer* expired, without carrying out registration when the UE returned to the old zone, such as in the 2Z system. On average, the total cost of the 2Z_Timer system was 10.6% lower than that of the 1Z system and 28.6% lower than that of the 2Z system.

The total cost according to the change in the probability ($q$) of returning to the previous zone is shown in Figure 9 when $T_h$ = 20 min and CMR = 1. For the 1Z system, the total cost was determined regardless of the $q$. In the 2Z system, as the $q$ increased, the UE frequently returned to the previous zone. The total cost was then decreased because the registration cost was reduced. For the same reason, the registration costs could be reduced in the 2Z_Timer system. However, compared to the 2Z system, its reduction effect was relatively small due to the registration cost caused by the *Timer* expiration. In this example, it was observed that the total cost of the 2Z_Timer system was always less than those of the 1Z and 2Z systems. The total cost of the 2Z_Timer system was decreased by 10.7% compared to that of the 1Z system and by 31.1% compared to that of the 2Z system.

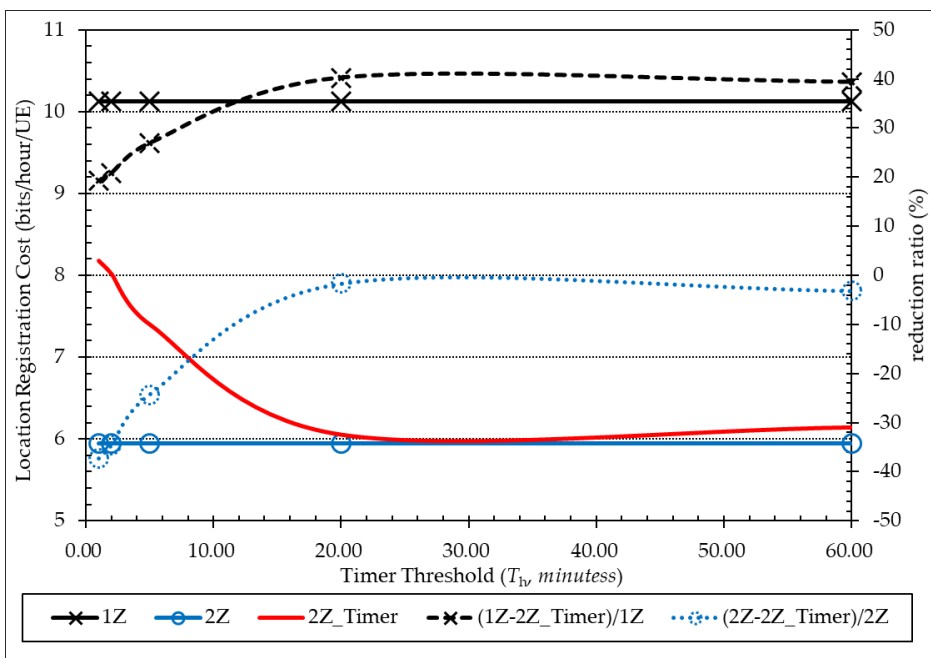

**Figure 6.** Location registration cost versus $T_h$.

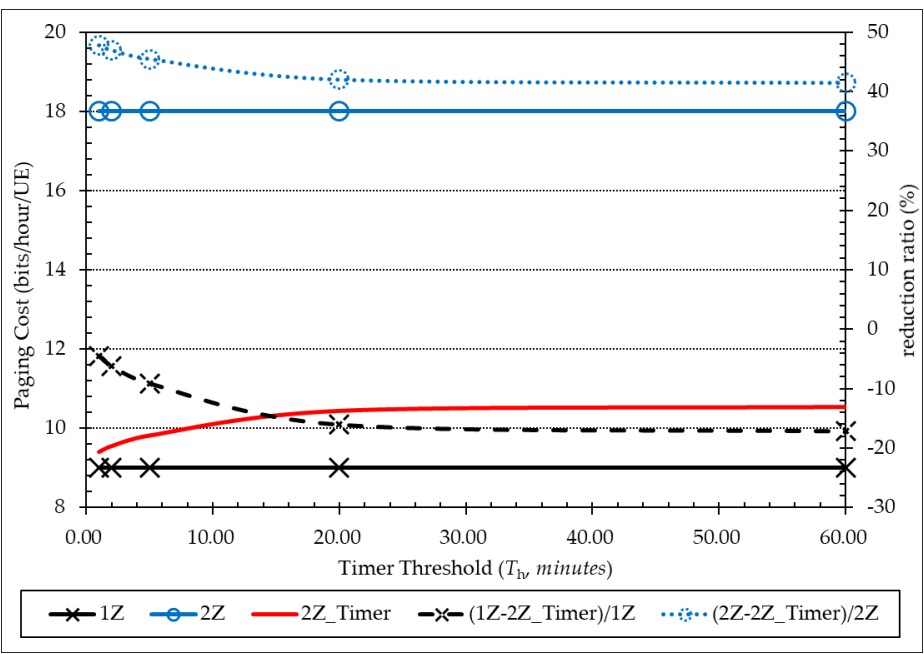

**Figure 7.** Paging cost versus $T_h$.

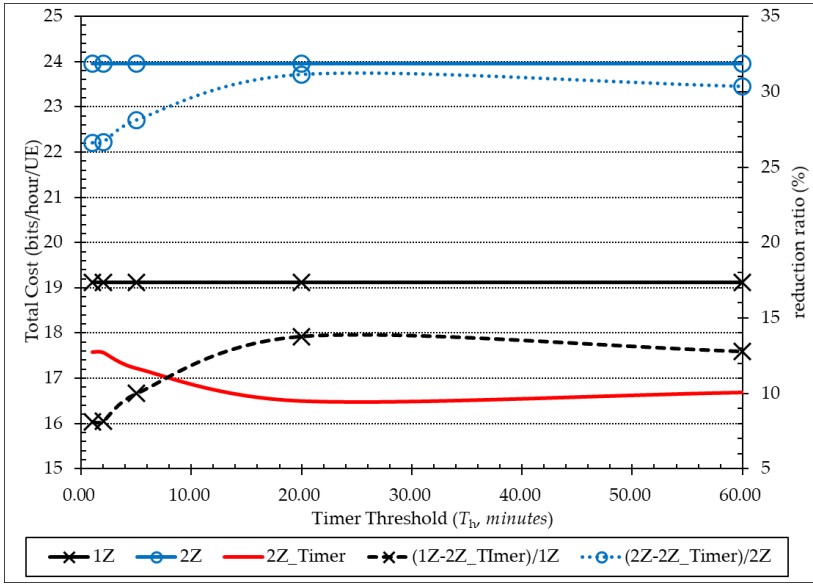

**Figure 8.** Total cost versus $T_h$.

Figure 10 shows the total costs versus the *n* at $T_h$ = 20 min and *q* = 0.4. The results show the total cost for *n* = 4, 9, and 16. Note that if the *n* increased, causing the number of cells that needed to be paged to increase, the performance of the 2Z_Timer system resulted in a reduction in the paging cost, further enhancing its performance compared to the other systems. The total cost of the 2Z_Timer system was decreased by an average of 30.1% compared to that of the 2Z system and by 13.4% compared to that of the 1Z system.

Table 3 summarizes the main results shown in Figures 5–10. As a result of analyzing various parameters, it was observed that the 2Z_Timer system always outperformed the 1Z and 2Z systems. On average, the total cost of the 2Z_Timer system was reduced by 10.6~15% compared to the 1Z system and by 27.7~31.1% compared to the 2Z system.

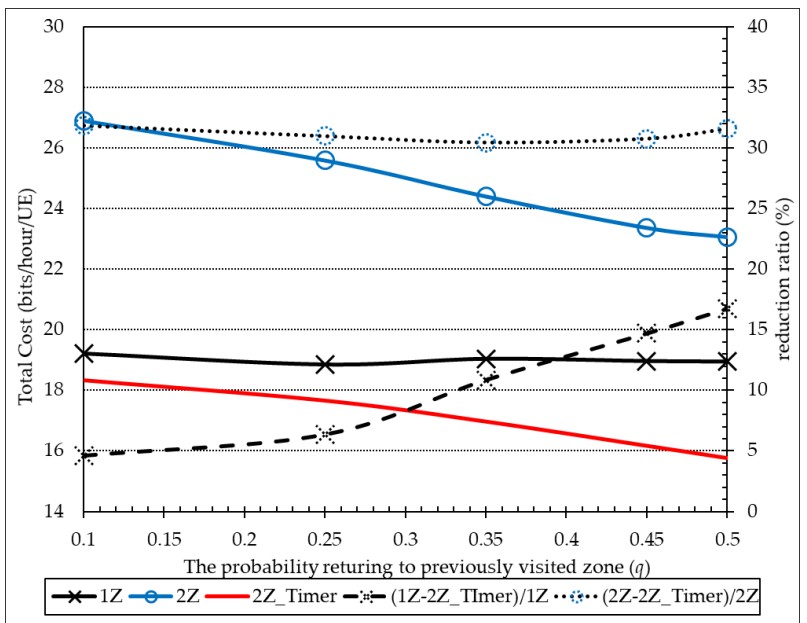

**Figure 9.** Total cost versus *q*.

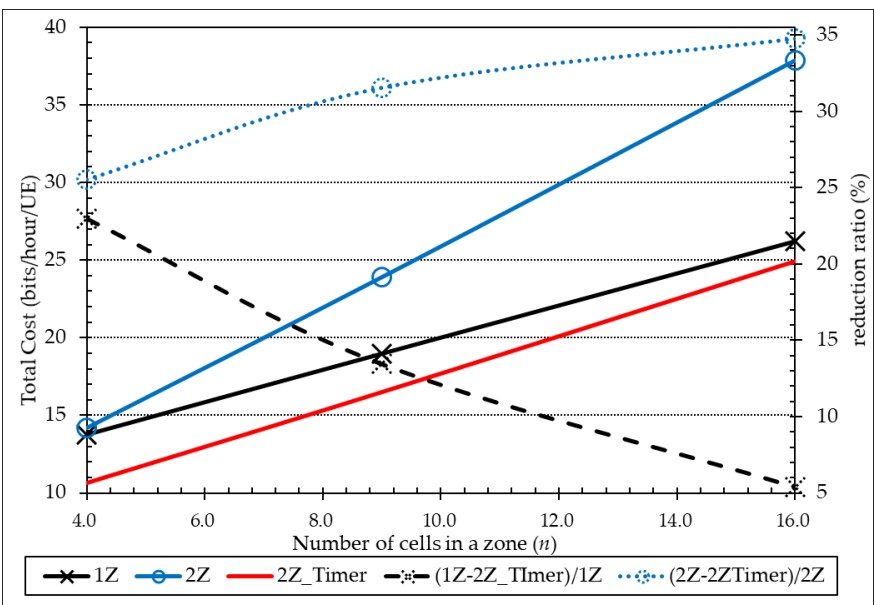

**Figure 10.** Total cost versus *n*.

Comparing the 1Z and 2Z systems, the performance of the 1Z system was always better than that of the 2Z system, except when the CMR was small at 0.1 for a paging delay of 1. This was because, in the case of the 2Z system, even if the UE returned to the previous RA, the paging costs were always twice those of the 1Z system. When the CMR was small, the dwell time in the RA was reduced, which often returned to the previous RA, eventually improving the performance of the 2Z system due to the reduction in the registration cost. Note that if multiple paging attempts are possible, the performance of the 2Z system can be superior to that of the 1Z system, as indicated in other studies [10–14].

Based on the numerical results, several key conclusions can be drawn, as follows:

(i) When an appropriate $T_h$ is set in consideration of the UE's CMR, the performance of the 2Z_Timer system is always superior to those of the 1Z and 2Z systems;

(ii) The performance of the 2Z_Timer system improves as the $T_h$ increases, although its improvement effect is not very significant at a certain threshold or higher;

(iii)   The total cost of the 2Z_Timer system decreases as the CMR (or $E[T_m]$) increases due to the reduced paging cost obtained by preventing the UE from entering other zones as the CMR increases;

(iv)   As the probability ($q$) of returning to the previous zone increases, the performance of the 2Z_Timer system is better than that of the 1Z system. Note that as the $q$ increases, the cost of the 2Z_Timer system decreases, similar to the total cost of the 2Z system;

(v)   If the paging delay is constrained by 1, the performance of the 1Z system is superior to that of the 2Z system. Nevertheless, if the CMR is relatively small, the total cost of the 2Z system can be less than that of the 1Z system. This is because, when the CMR is small in the 2Z system, the zone residence time is small, and the number of cases returning to the previous zone increases, resulting in a reduction in the registration cost;

(vi)   As the number of cells included in a zone increases, the performance of the 2Z_Timer system is further improved and is better than that of the 2Z system due to the reduced paging cost. Even when $n = 4$ is small, the 2Z_Timer system is always superior.

**Table 3.** Summary of main results (▼: total cost reduction ratio (%)).

| Call-to-mobility ratio (CMR) | *Timer* threshold ($T_h$) |
| --- | --- |
| 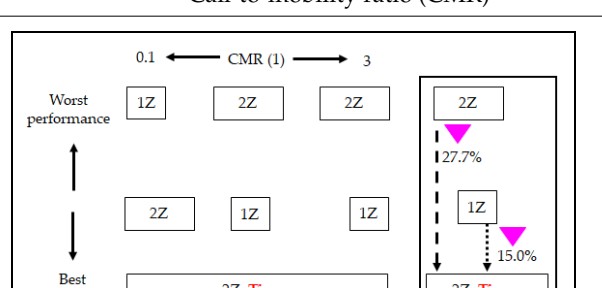 | 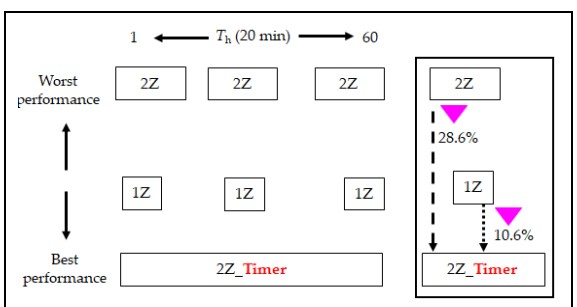 |
| 2Z performs better than 1Z when CMR is small; 1Z performs better than 2Z when CMR is large. | Performances of 1Z and 2Z do not change with $T_h$. For CMR = 1, $T_h$ = 20 min is appropriate. |
| The probability of returning to the previous RA ($q$) | Number of cells in an RA ($n$) |
| 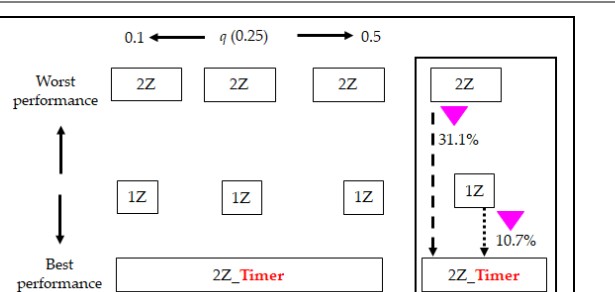 | 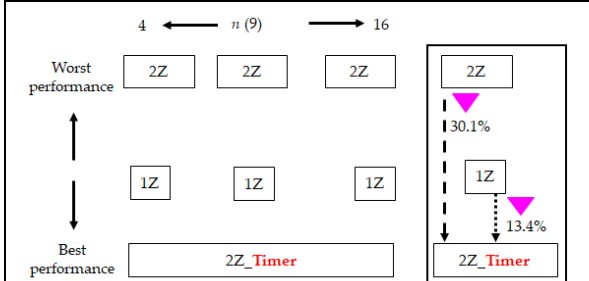 |
| 1Z performance is independent of $q$. As $q$ increases, the difference in performance between 1Z and 2Z becomes smaller. | Higher $n$ indicates better performance of 1Z than 2Z. 2Z_Timer's performance improves more than that of 2Z as $n$ increases. |

To summarize our results, a 2Z system was proposed to reduce the registration cost caused by the ping-pong effect, the most significant problem in the 1Z system. However, because the two RAs are managed in the database, the paging costs are twice as much as those of the 1Z system when the incoming call is connected. In contrast, in the 2Z_Timer system, the paging costs could be effectively reduced by driving a *Timer* for the previously visited RA when the UE enters a new RA, and the advantages (reduction in the registration cost at RA boundaries) of the 2Z system can be used. As a result of the performance comparison, it was observed that the proposed 2Z_Timer system always performed better than the 1Z and 2Z systems.

## 5. Conclusions

In this study, a new scheme of utilizing *Timer* for previously visited zones is proposed to reduce the paging cost with a two-zone registration method. The 1Z system currently used in wireless communication networks has one zone per registration area (RA) with registration traffic that frequently occurs at the RA boundary, which is considered the greatest disadvantage. To overcome these shortcomings, a 2Z system that can manage two zones as one RA has been proposed. However, this is also difficult to implement due to the load of the paging cost for the user equipment (UE) that has returned to the previous zone. Therefore, in the new 2Z_Timer system, the *Timer* for the previous zone is run for the UE entering a new RA so that the old zone's information is available only for a certain threshold ($T_\mathrm{h}$). In the 2Z_Timer system, the paging cost incurred is the same as that in the 2Z system if the *Timer* did not expire after returning to the previous zone. However, the paging cost can be reduced through a new registration process when entering a new zone, and when the *Timer* expires in the previous zone.

To evaluate the performance of the proposed system, a visual-flowchart-based RAPTOR software program was used to obtain the registration and paging costs for a total of 10,000 units of UE. First, if the call-to-mobility ratio (CMR) increased and the UE stayed in the zone for a long time, the performance of the 2Z system was worse than that of the 1Z system because the UE did not return to the previous zone. However, in the 2Z_Timer system, even if the CMR increased, the paging costs could be reduced when the UE stayed in the new zone, resulting in a decrease in the total cost. Second, as the *Timer* threshold ($T_\mathrm{h}$) increased, the performance of the 2Z_Timer system improved. Due to additional registration costs caused by UEs returning to the previous zone upon *Timer* expiration, the registration cost increased compared to that of the 2Z system, although it was lower than the registration cost of the 1Z system. In the 2Z_Timer system, the paging cost was always lower than that of the 2Z system. Although the paging cost of the 2Z_Timer system was slightly higher than the paging cost of the 1Z system, it remained at a similar level. Therefore, when the total costs of registration and paging were compared, it was ultimately confirmed that the 2Z_Timer system resulted in a cost reduction of 10.6% compared to the 1Z system and of 28.6% compared to the 2Z system. Third, as the probability of returning to the previous zone increased, the advantage of the 2Z system could be used in the 2Z_Timer system, which could reduce the registration cost. Even if the registration cost was increased due to *Timer* expiration, the performance of the 2Z_Timer system was superior to those of the 2Z and 1Z systems due to the reduction in the paging cost. Finally, in the case of paging all at once, as the number of cells included in a zone increased, the performance of the 2Z_Timer system was further improved due to the reduction in the paging cost. In summary, the newly proposed 2Z_Timer system outperformed the 1Z and 2Z systems as a result of simulating various system parameters. On average, the total (registration + paging) cost can be reduced from 10.6% to 15% compared to the 1Z system and from 27.7% to 31.1% compared to the 2Z system.

In this study, a simulation method was used to evaluate the performance of the 2Z_Timer system. However, this simulation environment could be very different from an actual wireless communication network. As a major system constraint, for the convenience of analysis, the same number of cells in a square-shaped RA was first assumed. However, in reality, the cell's size and the number of cells in an RA may differ by service region considering the radio propagation network. Second, as mentioned earlier, most wireless communication networks use the 1Z system due to its convenience of implementation. The issue of the difficulty of realizing a system with a *Timer* remains. Finally, if the paging delay is not 1, but multiple paging attempts are possible, the 2Z's performance may be better, as can be seen in other studies. For further studies, after deriving analytical results for the 2Z_Timer system, the simulation results should be confirmed using the actual wireless communication parameters. The performance of a mobility management system is influenced by various parameters. In an era when various mobility devices are expected to appear in the future, optimal values of the parameters for each device are required. To

obtain them, it is necessary to apply a machine learning algorithm that has recently been used in various fields.

In particular, new mobility management schemes using artificial intelligence machine learning algorithms (the so-called AI-powered zone-based registration approach) have been proposed. In this study, there was a limit to the application of an AI-powered method due to the difficulty of collecting training data on the moving characteristics of UE. In future studies, if various AI-powered algorithms are applied using a UE-specific CMR and $q$, better performance algorithms can be developed.

In the future of beyond 5G and 6G wireless communication services, a wide variety of mobile devices, including smartphones, are expected to emerge. Given the limited wireless frequency resources, mobility management functions for these devices will become increasingly important. Therefore, mobility management systems that can leverage the mobility characteristics (CMR and $q$) of each device will be necessary. Future research should focus on dynamic mobility management strategies tailored to the mobility characteristics of each device. Additionally, research should explore the use of a *Timer* in the 3Z system, which can offer a performance superior to that of the 2Z system.

**Author Contributions:** Modeling and data analysis: H.-S.J. and J.-H.B.; writing—original draft: H.-S.J.; writing—review and editing: J.-H.B. All authors have read and agreed to the published version of the manuscript.

**Funding:** This work was supported by grants (No. 2021R1F1A1049933 and No. 2022R1F1A1074141) from the National Research Foundation (NRF) funded by the Ministry of Science and ICT (MSIT), the Republic of Korea.

**Data Availability Statement:** Data is contained within the article.

**Acknowledgments:** The authors sincerely thank the anonymous reviewers for their hard work and efforts reviewing this paper.

**Conflicts of Interest:** The authors declare no conflicts of interest.

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
