# Peer review of "Performance Analysis of Two-Zone-Based Registration System with Timer in Wireless Communication Networks"

_electronics, doi:10.3390/electronics13010160_

Round 1

Reviewer 1 Report

Comments and Suggestions for Authors

The contribution is minimal. The paper is not organized well. More work on the organization is required as the proposed strategy, using a timer for the 2Z system, is introduced within the background (section 2. ).

The first two subsections of section 2 introduced the zone-based scheme. However, related works within this category are not presented.   

A detailed description of the proposed system and how it works differently from other existing works is required.

The Conclusion section should summarize the findings, not present the scheme.

Sections and subsections are not orderly numbered. 

Comments on the Quality of English Language

no comments

Author Response

We would like to thank you for your elaborate review and useful comments.

We have revised the paper according to your helpful comments despite the short revision time of 10 days requested, and replies are summarized in an attached file.

Considering your helpful comments, we have done our best to make the paper more significant.

In addition, English was corrected by a specialized English correction institution (HARRISCO, https://en.harrisco.net/).

Thank you very much again for your elaborate review and good response.

Reviewer 2 Report

Comments and Suggestions for Authors

The authors of this paper propose a 2Z system for managing two zones with a single registration area. My reviews are included below.

1.The introduction part is too short, Include a general diagram in the introduction part for a better understanding of the readers.

2.Include one paragraph describing the impact of the AI-powered zone-based registration approach and explain why AI is not employed.

3.Why is RAPTOR used for simulation?

4.Include a table with the results from Figs. 5 to 10 and explain briefly.

5.In the simulation model section, the system parameters for simulations must be referenced.

6.In the conclusion, explicitly define the system constraints.

7.Add one paragraph before the conclusion explaining the advantage of the proposed 2Z versus 1Z.

Comments on the Quality of English Language

Minor editing of English language required

Author Response

(The authors gave the same response as above.)

Reviewer 3 Report

Comments and Suggestions for Authors

This work introduces a two-zone registration system with a timer for cellular communication systems, which improves the system performance when there is a chance of returning to the previous zone within that period of time.

Although the manuscript is well written and results look very good, I do not recommend publication in its current form due to the following reasons:

- The literature review must be extended and updated. Recent papers from high rank journals (like those from IEEE) should be included. If there are no such papers, then it is a sign that this subject is not of great interest nowadays to the academic community...

- Still on the literature review, in the current version the authors mention 17 related papers in a single paragraph. It would be great if they could discuss these related work in more detail, perhaps in a section or subsection, and present a table listing their main characteristics, so that it becomes easier for the reader to understand what are their flaws and what is this paper doing that is significantly novel.

- The numerical results section should include a comparison with some of the strategies discussed in the related work.

- The performance of the proposed method depends on some parameters that are related to the user. For instance, the call to mobility ratio depends on the user. So, how to optimize or learn the optimum system parameters on the fly?

Author Response

(The authors gave the same response as above.)

Round 2

Reviewer 1 Report

Comments and Suggestions for Authors

There exist some typos that need correction such as "East to implement" in the advantage column Table 1.

What is the category of "so on [20-23]? second paragraph in the introduction.

Why the references in Table do not include the mentioned reference in the paragraph?

The revised manuscript still does not introduce the related work within the Zone-Based Registration Scheme. It only mentioned the references.

Comments on the Quality of English Language

Further proofreading is advised

Author Response

We would like to thank reviewer for your elaborate works and useful comments.

We have revised the paper according to your helpful comments, and replies are summarized as attached file.

Thank you again for your elaborate review and good response.

Truly yours.

Authors

Reviewer 3 Report

Comments and Suggestions for Authors

I thank the authors for the detailed answers to my comments. They made a great job in Tables I and II.

Author Response

Dear anonymous Reviewer:

We would like to thank you for your elaborate works and useful comments.

Truly yours.

Authors
